# Atropine Is a Suppressor of Epithelial–Mesenchymal Transition (EMT) That Reduces Stemness in Drug-Resistant Breast Cancer Cells

**DOI:** 10.3390/ijms23179849

**Published:** 2022-08-30

**Authors:** Emad A. Ahmed, Mayyadah A. Alkuwayti, Hairul-Islam M. Ibrahim

**Affiliations:** 1Biological Sciences Department, College of Science, King Faisal University, Al Ahsa 31982, Saudi Arabia; 2Lab of Molecular Physiology, Zoology Department, Faculty of Science, Assiut University, Assiut 71515, Egypt; 3Pondicherry Centre for Biological Science and Educational Trust, Pondicherry 605004, India

**Keywords:** atropine, epithelial–mesenchymal transition, stemness, drug-resistant breast cancer cells

## Abstract

Atropine (ATR) is extracted from a belladonna plant that belongs to a class of anticholinergic drugs and is therefore involved in the treatment of the overdose of cholinergic drugs or mushroom poisoning. It is a well-known blocker of muscarinic acetylcholine receptors (mAChRs) that are expressed in various tumor cells, including breast tumors from animal and human origin, but it has yet to be recommended as an anticancer drug. Our in silico docking analysis indicates that atropine has a roust virtual binding, with a stable binding energy, to two major signaling molecules involved in EMT regulation: E-cad and ZEB-2. For both, the gene and the protein expression level results show that atropine is an effective molecule in reducing epithelial–mesenchymal transition (EMT) and colony formation induced by TGF-B or carboplatin in both the mesenchymal-like cell line MDA-MB-231 and the epithelial-like cell line T47D. We conclude that atropine as a potential suppressor of EMT could be co-administrated with other chemotherapeutic drugs to reduce stemness in drug-resistant breast tumor cells.

## 1. Introduction

Atropine, a naturally occurring alkaloid from the Atropa belladonna plant, is used as anticholinergic medication to slow heart rate and to treat certain types of nerve agent and pesticide poisonings [1]. Atropine works as a competitive antagonist for the muscarinic acetylcholine receptors (mAChRs) and inhibits the parasympathetic nervous system via blocking of these receptors [2,3]. Besides being expressed in neural cells, mAChRs are expressed in various tumor cells, including breast cancer ones from animal and human origins; however, normal breast cells do not express these receptors [4]. The activation of mAChRs triggers cell proliferation, migration, invasion, and metastasis, making it a useful target for metronomic chemotherapy. These cancer processes, and other processes such as epithelial–mesenchymal transition (EMT), are mediated by the overexpression of AChRs in different kinds of tumors, including breast cancer [5].

The EMT is a reversible physiological process involved in cancer progression via the morphological repression of epithelial markers (e.g., E-cadherin, claudins, occludins) and acquiring of mesenchymal markers (e.g., vimentin, fibronectin, *N*-cadherin) [6]. On the other hand, tumors may acquire the stemness features of cancer cells and promote cancer cells maintenance, differentiation, and survival, which is dependent on the organization of stemness-supporting microenvironments [7,8,9]. For instance, the secretion of inflammatory cytokines creates an inflammatory microenvironment that may stimulate EMT and then lead to the generation of cancer stem cells (CSCs) [6,7].

Studies have shown a paradoxical and complex role for the transforming growth factor-β (TGF-β) during tumorigenesis due to its ability to switch from a tumor suppressor in normal and early stages of cancer to a tumor promoter in late stages [10,11]. Along this line, in claudin-low and metaplastic mammary tumors, TGF-β can promote tumor progression by supporting the oncogenic induction of EMT [12,13]. Tumors may develop resistance to chemotherapy where small populations of cells, the CSCs, acquire stemness properties, which promote cancer progression and enhance self-renewal, cloning, growing, metastasizing, and reproliferating capacities.

Metastatic development and resistance to chemotherapy are major therapeutic obstacles of breast cancer [14,15]. As an example, the cisplatin derivative, carboplatin, that is widely used in the treatment of several tumor types, can unfortunately enhance drug resistance in some patients with biochemical mechanisms that are precisely unknown [16]. The chemo-resistant triple-negative (ER-/PR-/HER2-) breast cancer MDA-MB-231 cell line was found to display greater CSCs characteristics than other breast cancers; therefore, it is more aggressive. This breast cancer cell line is a claudin-low mesenchymal-like and basal B cell line; however, for other cancer cell lines subtypes, T47D is a luminal A (ER+/PR+/HER2-) epithelial-like cell line [17]. On the other hand, a considerable number of in vitro and in vivo studies have indicated that vitamin D is used in combination with other chemotherapeutic drugs as an efficient biomolecule that reduces drug resistance and EMT in cancer cell lines [18,19].

Interestingly, although several recent studies have exhibited muscarinic receptors as new targets for breast tumor therapy, and atropine is widely used as routine medication treating overdose of cholinergic drugs, atropine biological anti-cancer activities have been reported by only a few studies [3,4,5,20]. In addition, not much attention has been given to the role of atropine in reducing the stemness and the EMT characteristics, even though it is a well-known anti-AChRs drug with a higher IC50.

In the present study, we aim to investigate the role of atropine in reducing EMT and stemness in vitro using two breast cancer cell lines, the mesenchymal-like cell line MDA-MB-231, and the epithelial-like cell line T47D. Our results suggest that atropine could be used in combination with other chemotherapeutic agents to challenge the drug resistance development and to reduce EMT as well as stemness in breast cancer cells.

## 2. Results

### 2.1. Atropine Interacts with E-cad and Zeb-2 In Silico

The potential interaction of atropine with E-cad and ZEB-2 was analyzed using a molecular docking approach. Binding energy, legend bonds, and docked amino acids residues are shown in Figure 1, Table 1. Atropine was found to interact with E-cad and a conventional hydrogen bond (C12-H) is formed at chain A (GLN OE1) with bond length 2.01 Å and a binding energy of 6.75 kcal/mol (Figure 1A). In addition, atropine binds with ZEB-2, with a binding energy of 4.78 kcal/mol at conventional hydrogen bond C12-H (chain A: VAL’46’O’ 1.75 Å), and at carbon hydrogen bond interaction C8 and C12-O (Chain A: PRO’10’O’, 3.09 Å, and Chain A: LYS’49’CE’, 3.72 Å, respectively) (Figure 1B). These results indicate that atropine has roust and stable binding energy with two major signaling molecules involved in EMT regulation.

### 2.2. Atropine Reduces the Proliferation Rate in Breast Cancer Cell Lines

The cytotoxic effect of various doses of atropine (0–50 µM) was evaluated in normal breast cell line and in breast cancer MDA-MB-231 and T47D cell lines (Figure 2A–C). The cell viability was quantified after incubating the cells with atropine for 48 h for normal cells and at time points 24, 48, and 72 h for breast cancer cells. The IC_50_ value of atropine was more than 60 μM for normal breast cell lines but was around 20 μM for other cell lines. Representative images of the cytotoxic effect of atropine on MDA-MB-231 are shown in Figure 2D. The results show that atropine significantly reduces the viability in cancer cell lines (at 10 μM concentrations and above) relative to the normal cell lines. These results indicate that atropine is a genotoxic drug that reduces cell viability in the breast cancer cell lines MDA-MB-231 and T47D.

### 2.3. Atropine Suppresses EMT in MDA-MB-231 and T47D Cancer Cell Lines

Next, we supplemented TGF-B treated MDA-MB-231 and T47D cell lines with atropine (5 and 10 μmol) for 48 h. Then, the levels of mesenchymal and cancer stem-cell-based mRNA expression were analyzed by qPCR. Data showed an inhibitory effect of atropine on TGF-B mediating EMT in both cell lines through the reduction of the expression of CD-44 and C- Myc (Figure 3A,B). While the E-cad expression level was higher, and CD-44 and C- Myc levels were lower in DMSO-treated breast cancer cells, TGF-B treatment reduced E-cad levels in both cell lines, and the reduction was insignificant in the mesenchymal-like MDA-MB-231. In the meantime, TGF-B treatment induced a significant increase in the levels of CD-44 and C- Myc in both cell lines. These results were confirmed via immunoblot analysis where E-cad protein levels increased significantly in TGF-B-treated cell lines after being treated with atropine. In addition, CD-44 and c-Myc were downregulated in TGF-B/atropine-treated cells relative to the TGF-B treated ones in both cancer cell lines (Figure 3C). These results clarify the fact that atropine reduces EMT in both the chemo-resistant, triple-negative, claudin-low mesenchymal-like breast cancer (MDA-MB-231) cell line and the luminal A epithelial-like cell line (T47D). Atropine action was more efficient on the later cell line.

### 2.4. Atropine Abrogates the Cancer Stem Cell Modifications and Suppresses Migration and Invasion in MDA-MB-231 and T47D Cell Lines

The phase-contrast microscopic images of control TGF-β and TGF-β as well as atropine-treated cells were analyzed to study the effect of atropine treatment on the cellular morphological modifications in cancer cell lines (Figure 4A,B). Cytological investigations revealed that atropine suppresses EMT, as shown, by increasing the extent of membrane blebbing (arrows) and disturbing the cytoskeleton of mesenchymal-like cells induced after TGF-β treatment (Figure 4A). Therefore, atropine treatment appears to reduce mesenchymal modifications that have the characteristics of a fibroblast-like shape.

Next, we investigated the inhibitory effect of atropine on the migration and invasion of breast cancer cells. The wound-healing assay was performed to test the impact of atropine on retarding the migration activity of both cancer cell lines. DMSO-treated MDA-MB-231 and T47D cells displayed significant migration, while wound closure by cells treated with 10 μM atropine showed a delay in wound closure in both cell lines (Figure 4C,D). These data show that atropine reduces the migratory activity of breast cancer cells. Then, we estimated the ability of atropine to inhibit MDA-MB-231 and T47D cell invasion by performing a Matrigel invasion assay. The atropine treatment, at a concentration of 10 μM, significantly inhibited MDA-MB-231 and T47D cell invasion relative to the DMSO-treated cells (Figure 4E,F). These findings indicate that atropine has anti-invasive effects on MDA-MB-231 and T47D cells.

### 2.5. The Expression of Breast Cancer Mesenchymal Modification Genes Is Dysregulated after Atropine Treatment

Studies have indicated that vitamin D could be used in combination with other chemotherapeutic drugs as an efficient biomolecule to reduce drug resistance and EMT [21]. Therefore, we checked the repression effect of atropine on EMT inducers as compared with the effect of vit D. The MDA-MB-231 and T47D breast cancer cell lines were cultured in the presence of atropine (25 μmol/L) or DMSO with or without TGF-β (10 ng/mL) induction and Vit-D (10 nM)-supplemented cells (Figure 5). As shown in Figure 5B, atropine diminishes the action of TGF-β-inducing EMT in the mesenchymal-like BC cells MDA-MB231 via downregulating ZEB-2. In the epithelial-like T47D cell line, TGF-β enhanced EMT via the reduction of the epithelial cells marker E-cad and increasing the ZEB-2 level; however, atropine or vit D significantly modulated the action of TGF-β and then reduced EM (Figure 5C). Atropine seems to enhance the induction of a similar expression level of E-cad to what is induced after vit D treatment. In the meantime, atropine reduced the protein levels of CD-44 and c-Myc, upregulated after treating both cancer cell lines by TGF-β (Figure 5D,E). This was more efficient as compared with vit D. In parallel with that, atropine retarded EMT in both cell lines through the inhibition of one of the EMT-promoting molecules, Sox2 (Figure 5E,F). These results reflect a possible role for atropine in EMT regulation when used alone or in combination with other drugs.

### 2.6. Atropine Abrogates the Carboplatin-Induced Cancer Stem Cell Modifications in Breast Cancer Cell Lines

To further confirm that atropine is able to reduce EMT induced by drug-resistant cells, we used carboplatin as a model to enhance cell migration and invasion activity. As shown in Figure 6, cells were treated with vit D, atropine, or both, and showed significant reduction in migration activity in both cell lines, and relative to the carboplatin-treated cells, the inhibitory effect was more efficient in the mesenchymal-like triple-negative cell line (Figure 6A,B). In the meantime, invasion activity enhanced by carboplatin was found to be reduced in both cell lines, and the reduction was significant in triple-negative cell lines (Figure 6C,D).

### 2.7. Atropine Alleviates Carboplatin-Induced Mesenchymal Transition and Stemness in Breast Cancer Cells

Next, we checked the colony formation in both BC cell lines after being treated with carboplatin only or with carboplatin in combination with vit D or atropine. As shown in the representative images in Figure 7A, the cells were treated with atropine and vit D to challenge carboplatin-inducing colony formation in MDA-MB-231 and T47D cell lines, and atropine decreased the size of the colonies and reduced the number of cells per colonies. To further check the inhibitory effect of atropine in EMT in the carboplatin model, the treated cell lines were quantified using Western immune blot analysis of E-cad, CD-44, and c-Myc. Atropine reduced the action of carboplatin-inducing EMT through the upregulation of E-cad and the downregulation of CD-44 and c-Myc in both the mesenchymal-like MDA-MB231 cell lines and the epithelial-like T47D cell lines. Together, our data confirmed that atropine is involved in EMT regulation and can reduce the colony formation of metastatic breast cancer cells and then alleviate stemness.

## 3. Discussion

In the current study, we found that atropine is a potential suppressor of EMT that could be co-administrated along with other chemotherapeutic drugs to prevent stemness in drug-resistant breast tumor cells. Our results show that atropine may bind to E-cad and ZEB-2, thus helping to reduce the breast cancer metastatic development and resistance to chemotherapy in two cell lines from different origins, the aggressive claudin-low mesenchymal-like cells, and the T47D cells, which is a luminal A epithelial-like cell line.

Since both TGF-β and carboplatin can induce EMT via epithelial–mesenchymal transition, we used both to induce EMT in breast cancer cell lines to confirm our findings. In fact, previous studies attributed the anticancer activity of atropine to its role as a competitive antagonist that blocks the mAChRs [2,3]. In this regard, M3-mAChRs were found to be involved in tumor cell migration, perineural invasion, and EMT during cholangiocarcinoma [22]. However, in a recent study, an anticancer activity for the alkaloid atropine, isolated from the hyoscyamus albus plant, was shown on different cancerous cell lines [23]. In that study, atropine was reported to have a strong cytotoxic activity against DU-145 (IC50 = 417 µg/mL) and on U-373 MG cells (IC50 = 894 µg/mL), but no more mechanistic details have been provided [23].

In the current model, atropine reduced EMT in both the chemo-resistant, triple-negative, claudin-low mesenchymal-like breast cancer (MDA-MB-231) cell line and the luminal A epithelial-like cell line (T47D) through the upregulation of the level of E-cad and the reduction of both CD-44 and c-Myc. These results agree with the recent finding of Yang and co-authors, who reported that the non-selective mAChR antagonist, atropine, significantly abrogated the decrease in E-cad expression and the increase in vimentin and α-SMA expression induced by TGF-β1 in alveolar epithelial cells (A549). However, the TGF-β1 effect was enhanced by the acetylcholinesterase inhibitor physostigmine [24]. The presence of mAChRs has been detected in different types of tumor cells, and these receptors were reported to be linked with tumorigenesis. MAChRs are expressed in murine and human mammary adenocarcinomas but are absent from normal mammary cells of the same origins [4,25]. In addition, mAChRs were demonstrated to be involved in breast cancer progression, pointing out to a main role for mAChRs as oncogenic receptors [25]. In the meantime, the treatment of cholangiocarcinoma with the M3-mAChR agonist pilocarpine or upregulation of M3-mAChR significantly promoted migration and perineural invasion. However, the M3-mAChR antagonist atropine blocked these effects [23]. The present study results indicate that atropine modulates E-cad levels and reduces the gene and the protein levels of CD-44 and c-Myc that are upregulated after treating both cancer cell lines by TGF-β or carboplatin. In parallel with that, the upregulated AKT, E-cadherin, vimentin, and Snail, which are components of the phosphatidylinositol 3-kinase/AKT signaling pathway and EMT, were reported to be blocked by atropine [23]. Interestingly, a higher expression level of mAChRs has been shown in breast tumors, which is not the case in normal breast cells and tissues [4,25,26]. The stimulation of mAChR with the carbachol (muscarinic agonist) was found to exhibit potential different steps of tumor progression in breast cancer cells and mice. However, it was observed that the selective M3 antagonists, p-F-HHSiD and atropine, can inhibit the effect of carbachol on LMM3 cell growth triggered by the activation of this receptor subtype [27].

A considerable number of in vitro and in vivo studies indicated that the combined treatment of vit D or vit D-derivatives with cytotoxic agents is more efficient in the reduction of the EMT process in various types of cancer lines [18,19,20,21]. Consistent with that, the loss of vit D receptors from human breast cancer cells was found to promote EMT in these cells [28]. Herein, atropine appears to induce a similar effect to what is induced by vit D. Moreover, the combined treatment of atropine and vit D is more efficient in the reduction of EMT and colony formation in both cell lines. The resistance caused after exposure to carboplatin was suggested to provide valuable clues for more effective rational drug design in platinum-based therapy and enable the development of new therapeutic strategies [29]. Therefore, in the current study, we show that atropine is involved in EMT regulation and can reduce the colony formation of metastatic breast cancer cells and then alleviate stemness.

## 4. Materials and Methods

### 4.1. In Silico Analysis

The potential interactions of atropine with ZEB-2 and E-cad were examined using docking analysis and Autodock tools (ADT) v1.5.4 and Autodock v4.2 program (http://www.scripps.edu/mb/olson/doc/autodock, accessed on 28 August 2021). The 3D chemical structure of ZEB-2 and E-cad were retrieved from the Protein Data Bank (http://www.pdb.org). The 3D structure of atropine was retrieved from the Pubchem compound database (http://www.ncbi.nlm.nih.gov/pccompound, accessed on 28 August 2021). The active sites of the target proteins were identified using Q-site Finder. Ligands docked to the receptor were reflected as a rigid body and receptors were considered as a flexible factor. The obtained results were scored and analyzed based on the predicted binding energy.

### 4.2. Cell Lines and Culture

The human breast cancer MDA-MB-231 and T47D cell lines and the normal breast cancer cell lines were cultured under standard conditions (5% CO_2_, 37 °C, 95% humidity) in Duelbecco’s minimal essential medium (DMEM) supplemented with penicillin–streptomycin–neomycin (1%), L-glutamine (2 mM), and heat-inactivated FBS (10%). Atropine (purity, ≥95%) was purchased from Sigma-Aldrich, dissolved in DMSO and added to medium depending on the required concentration. Normal cell lines were treated with various concentrations of atropine (1, 3, 6, 12, 25, 50, and 100 µM). On the other hand, breast cancer cells were treated with various concentrations of atropine (1, 5, 20, 50, and 100 µM) prior to being subjected to various assays. Negative control cells were treated with the standard culture medium and DMSO. After being harvested, cells were counted using a hemocytometer, and morphological changes were assessed by phase-contrast microscopy.

### 4.3. Cell Proliferation and MTT Assay

MTT assays were conducted to check effects of atropine on the proliferation rate of cancer and non-cancerous cell lines. Briefly, DMSO and atropine treated breast cancer MDA-MB-231 and T47D cell lines, and normal breast cell lines were cultured in DMEM supplemented with 10% FBS and antibiotics and incubated under standard conditions (5% CO_2_, 37 °C, 95% humidity). Cells (4 × 10^5^ cells per well) were seeded into 12-well cell culture dish, grown to 70% confluence, and then incubated for 12–24 h with atropine (1–100 μM). MTT reagent (20 μL) was added to treated cell lines and incubated for 4 h. Next, the media were aspirated, and the chromogenic colors were measured at 490 nm using a microplate reader. The effect of atropine on cell viability was evaluated as the proportion of treated cells relative to the viable vehicle-treated control cells and the arbitrarily deemed to be 100%. For EMT induction, cells were reseeded in a medium and incubated along with TGF-β (10 ng/mL) and atropine (at concentrations of 5 and 10 µM) for 24 h. Subsequent MTT procedures were performed as described above.

### 4.4. Cell Migration Assay (Wound-Healing Assay)

MDA -MB-231 and T47D were seeded (5 × 10^4^ cells/well) in 24-well plates in the serum-free DMEM culture medium. Cells were grown to 80% confluency, rinsed with phosphate-buffered saline (PBS), and then starved for 6 h in serum-free medium. A sterile 100-µL pipette tip was subsequently used to create wounds, after which all wells were washed with media to remove non-adherent cells. The cells were then treated with 0, 10, and 50 µM atropine. Images were captured with the inverted microscope at different time points (0, 12, 24, and 48 h) post-atropine supplementation. To monitor cell migration into the wounded region, images of the cultures were obtained by optical microscopy (magnification: 200×), and the area of the wounded region in each image was calculated with Image J software.

### 4.5. Cell-Invasion Assay (Transwell Assay)

Cell invasion was assessed using transwell cell culture Boyden chambers, according to the manufacturer’s protocol. Gelatin coated 12-well plate cell culture inserts (BD Biosciences) with a polyethylene terephthalate membrane (8-µm porosity) and the inserts were incubated for 6 h at 37 °C. Before atropine treatment, 100 µL of MDA cells (5 × 10^4^) were seeded in the upper chamber in serum-free media treated with atropine at the indicated concentrations and 700 µL of DMEM medium supplemented with 10% FBS was added to the lower chamber. The cells were then incubated for 24 h at 37 °C. Then, the remaining cells on the top surface of the membrane were removed using a cotton swab, and the cells on the bottom of the membrane were fixed in cold methanol (75%) for 15 min and washed with PBS three times. Afterward, the cells were stained with Giemsa (30%) staining solution and washed with PBS. Then, cells in five randomly selected fields were counted under a light microscope at 20× objective magnification. The number of migrated cells was tallied by optical microscopy (magnification: 200×) and manual counting. All assays were performed in triplicate.

### 4.6. RNA Isolation and Quantitative Real-Time PCR

TRIzol Reagent (Invitrogen, Waltham, MA, USA) was used to extract total RNA from the cells. cDNA was synthesized from 500 ng/gene of total RNA using a reverse transcriptase (TaKaRa, Kusatsu, Japan). RNA was reverse-transcribed to cDNA using miScript Reverse Transcription Kit (Qiagen, Hilden, Germany). RNA expression was measured by quantitative real-time PCR in VII 7A applied biosystems (Applied Biosystems, Waltham, MA, USA) using the Taqman gene probe (Table 2) (Thermo Scientific, Waltham, MA, USA) and SYBR-Green method (TaKaRa, Kusatsu, Japan) according to the manufacturer’s instructions. mRNA expression was calculated using internal control 2.

### 4.7. Western Blot Analysis

Treated cells were harvested for 24 h and then washed with PBS. Cells were lysed with RIPA lysis buffer and 1 × protease inhibitor cocktail, and then the lysate was centrifuged at 8000 rpm for 15 min to remove debris. The lysate supernatant was preserved at −80 °C. Protein concentrations were estimated by Bradford assay. The equivalent of 50 µg of protein extract was separated by SDS-PAGE and then transferred to polyvinylidene difluoride (PVDF) membranes (pore size: 0.45 µm, Bio-Rad, Hercules, CA, USA), which were treated with blocking buffer (5% non-fat dry milk) for 1 h at room temperature before being probed with the appropriate primary antibodies 1:1000 overnight at 4 °C, according to the manufacturer’s protocol. The membranes were then washed with TBST buffer and incubated with HRP-conjugated secondary anti-mouse/rabbit IgG antibodies (1:2000 Santa Cruz, CA, USA) for 1 h at room temperature. The blots were detected by an enhanced ECL chemiluminescence system (LICOR detection system) and then quantified by densitometry using Image J software. The primary antibodies included ZEB-2 mouse monoclonal antibody (1:750, Invitrogen, Waltham, MA, USA), E-cad rabbit polyclonal antibody (1:1500, Invitrogen, Waltham, MA, USA), β-Catenin rabbit polyclonal antibody (1:1000, Biorbyt, Cambridge, UK), CD-44 rabbit polyclonal antibody (1:500, Invitrogen, Waltham, MA, USA), c-Myc mouse monoclonal antibody (1:1000, Invitrogen, Waltham, MA, USA), SOX-2 mouse monoclonal antibody (1:1000 Invitrogen, Waltham, MA, USA), and β-actin rabbit polyclonal antibody (1:2000, Cell Signaling Technology, Beverly, MA, USA). For migration, invasion and Western blot analysis included carboplatin or vit D, monolayers of cell lines were treated with carboplatin only, carboplatin (5 nM) and Vit-D-β (10 nm), and carboplatin and atropine (25 nM). Subsequent procedures were conducted as described above.

### 4.8. Statistical Analysis

Data are expressed as mean ± SD. The significant differences between the DMSO-treated cells and atropine-treated cells were analyzed by the two-ways analyses of variance. The value of *p* < 0.05 and *p* < 0.01 were considered to be statistically significant and are represented by * and **, respectively.

## 5. Conclusions

Our findings highlight atropine as a potential suppressor of EMT that could be co-administrated along with other chemotherapeutic drugs to prevent stemness in drug-resistant breast tumor cells. Atropine may bind to EMT-signaling molecules such as E-cad and ZEB-2 or inhibit the breast cancer cells receptors and thus reduce the resistance to drugs and breast cancer metastatic development.

## Figures and Tables

**Figure 1 ijms-23-09849-f001:**
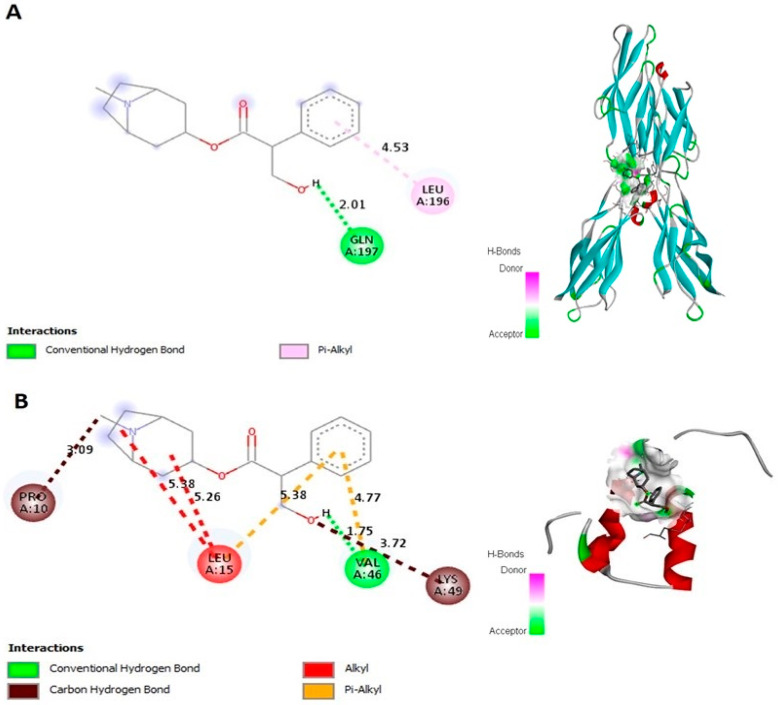
Molecular docking in silico docking of atropine (CID: 174174) and (**A**) Human E-cad (PDB-ID-1D5R) and (**B**) Human ZEB2 (PDB-ID-2D8V) proteins binding analysis. Computational binding was performed using AutoDock software to demonstrate the illustration of interactions in the hydrophobic bond and other polar bonds of E-cad and ZEB2. It shows the amino acid residue analysis of the interacted bond and its length, together with the binding pocket of ligand–receptor interactions.

**Figure 2 ijms-23-09849-f002:**
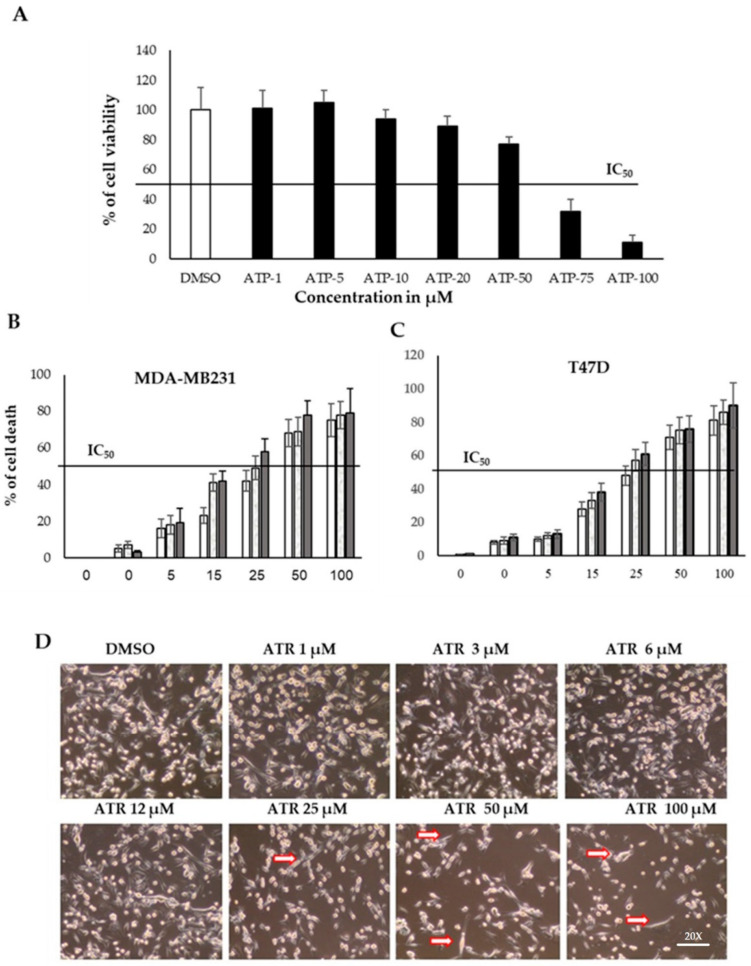
Atropine reduces the proliferation rate of breast cancer cell lines. Cells were supplemented with atropine (0–100 μmol) for different time periods, and cell viability was assessed with MTT assays. Normal breast cells, MDA-MB-231, and T47D were treated with TGF-β (10 ng/mL) or a vehicle control (0.1% DMSO). Cell viability was quantified after incubating cells with atropine for 48 h for normal cells (**A**), and at time points 24, 48, and 72 h for breast cancer cell lines (**B**,**C**). The IC_50_ value of atropine was more than 50 μM for normal breast cell lines but was less than 15 μM for other cell lines. (**D**) Representative images showing the epithelial modification in breast cancer cells, red arrows indicate fibroblast-like cellular structures. Results are expressed as the mean ± SD of triplicate measurements.

**Figure 3 ijms-23-09849-f003:**
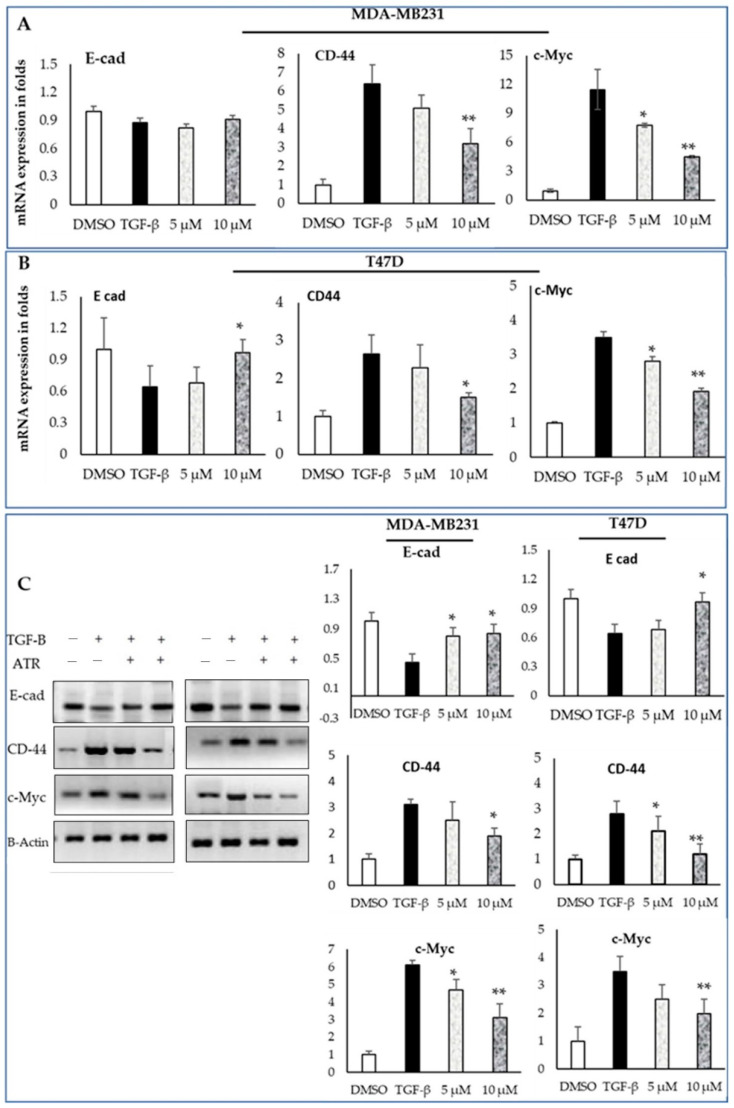
Atropine suppresses EMT in MDA-MB-231 and T47D cancer cell lines. (**A**,**B**) TGF-B-treated MDA-MB-231 and T47D cells were supplemented with atropine (5 and 10 μmol) for 48 h. The levels of mesenchymal and cancer stem-cell-based mRNA were analyzed by quantitative PCR. (**C**) TGF-B treated MDA-MB-231 and T47D cells were supplemented with atropine (5 and 10 μmol) for 48 h. The levels of mesenchymal and cancer stem-cell-based protein markers were analyzed by Western blotting. β-actin was utilized as the internal control. The results are expressed as the mean ± SD of triplicate measurements, * *p* < 0.05 and ** *p* < 0.01 when evaluated with control cells.

**Figure 4 ijms-23-09849-f004:**
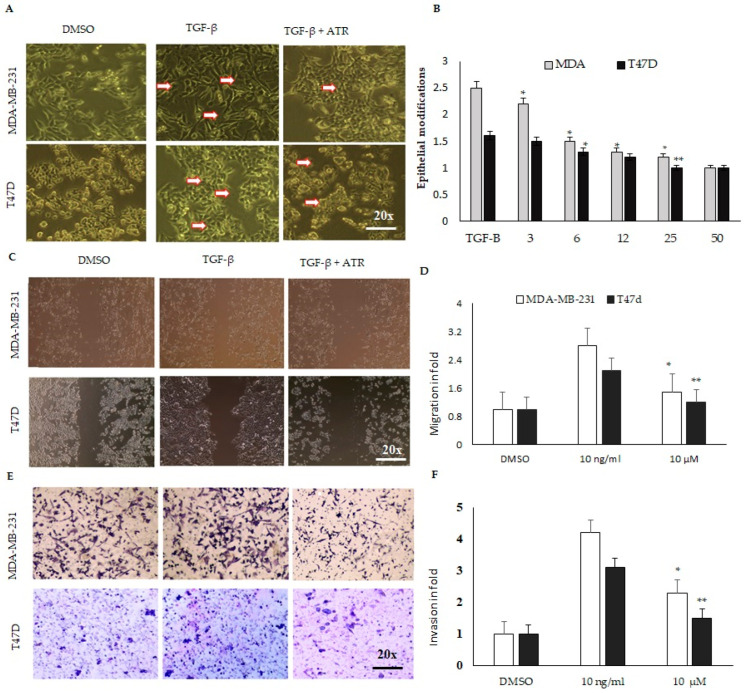
Atropine abrogates the cancer stem cell modifications in breast cancer cells. (**A**) MDA-MB-μ231 and T47D cells were treated with TGF-β challenged with atropine and the cellular structural modifications are shown. The phase-contrast microscopic images of control, TGF-β and TGF-β as well as atropine-treated cells. The red arrow mark indicates the mesenchymal modification from epithelial structure (TGF-β) and retained in atropine treatment. (**B**) MDA-MB-231 and T47D cell monolayers in 6-well plates were scratched down and wells were supplemented with TGF-β and TGF-β as well as atropine for 24 h. The migration was observed with an optical microscope (200× magnification) for recording wound-healing activity. (**C**) MDA-MB-231 and T47D cells treated with TGF-β and TGF-β as well as atropine to the upper chambers of Matrigel-coated transwells, and invasion was inspected via total cell counting in reversal plot of transwell. Cells crossed to the lower chamber after 24 h. The inhibition percentage of invasion was quantified and is expressed relative to the control (untreated cells), whose level of invasion was set at fold change. (**E**) Representative images of the invasion activity of treated cells. The migration (**D**) and the invasion (**F**) activity of breast cancer cell lines treated with DMSO, TGF-β (10 ng/mL) or TGF-β (10 ng/mL), and atropine 10 µM. Results are shown as mean ± SD from a representative experiment out of three independent experiments studied in quadruplicates and produced similar results. * *p* < 0.05 and ** *p* < 0.01 (one-way ANOVA); horizontal bars denote statistical comparisons.

**Figure 5 ijms-23-09849-f005:**
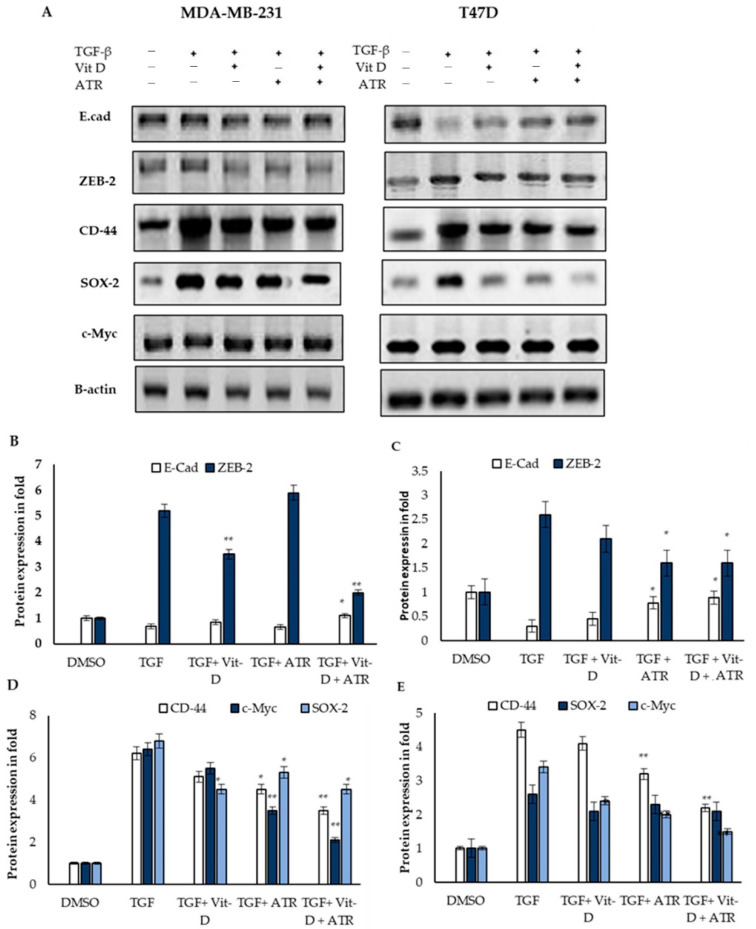
The expression of breast cancer mesenchymal modification proteins is modified after atropine treatment. The MDA-MB-231 and T47D breast cancer cell lines were cultured in the presence of DMSO or TGF-β (10 ng/mL), the latter having been treated with vit D (10 nM) or vit D and atropine. As a protein internal control, 3A β-actin (actin) was used. Representatives immunoblot (**A**) and protein quantification (**B**–**E**) of E-cad, ZEB-2, CD-44, SOX-2, and c-Myc were estimated after 48 h of treated conditions seen above. Results are shown as mean ± SD from a representative experiment out of three independent experiments studied in quadruplicates and produced similar results. * *p* < 0.05 and ** *p* < 0.01 (one-way ANOVA); horizontal bars denote statistical comparisons.

**Figure 6 ijms-23-09849-f006:**
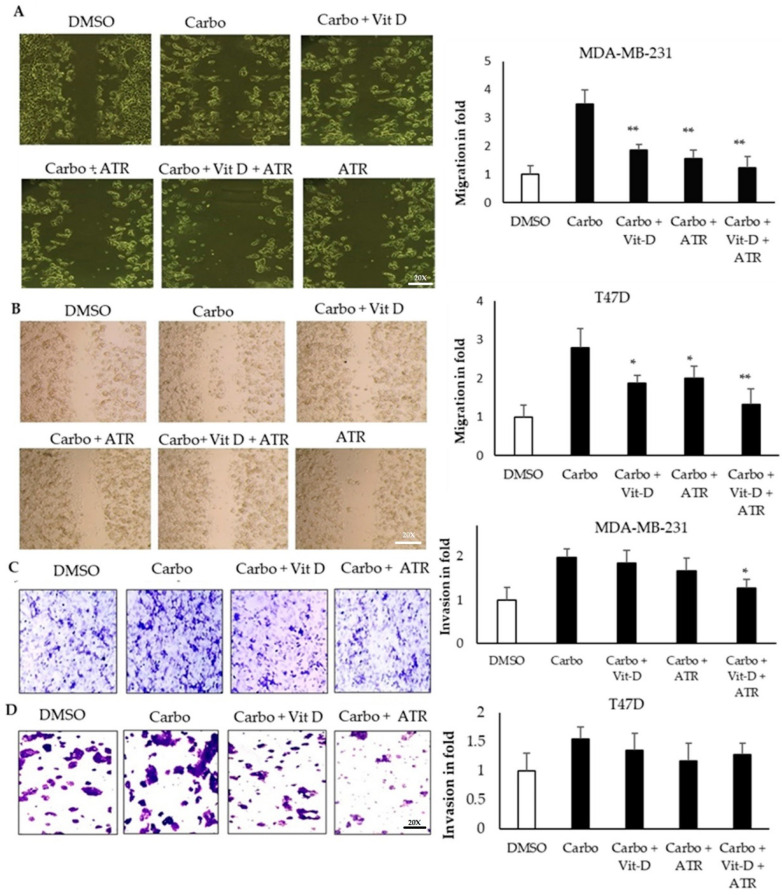
Atropine abrogates the carboplatin (cardo)-induced vit-D-supplemented cancer stem cell modifications in breast cancer cell lines. MDA-MB-231 and T47D cell monolayers were treated with carboplatin only, carboplatin (5 nM)-treated vit-D-β (10 nm), carboplatin and atropine (25 μmol/L) treatments, or carboplatin, vit D, and atropine. The migration activity (**A**,**B**) was assessed using EVOS optical microscope (200× magnification). (**A**) The phase-contrast microscopic images of treated MDA-MB-231 cell lines were recorded, right side. (**B**) Same recording for T47D for recording wound-healing activity. ImageJ tool was used to calculate the area of recovered wounds. (**C,D**) Invasion activity of MDA-MB-231 and T47D cells treated with control, carbo, carbo and vit D, carbo and atropine, or carbo and vit D, and atropine. Data are shown as mean ± SD from a representative experiment out of three independent experiments studied in quadruplicates and produced similar results. * *p* < 0.05 and ** *p* < 0.01 (one-way ANOVA); horizontal bars denote statistical comparisons.

**Figure 7 ijms-23-09849-f007:**
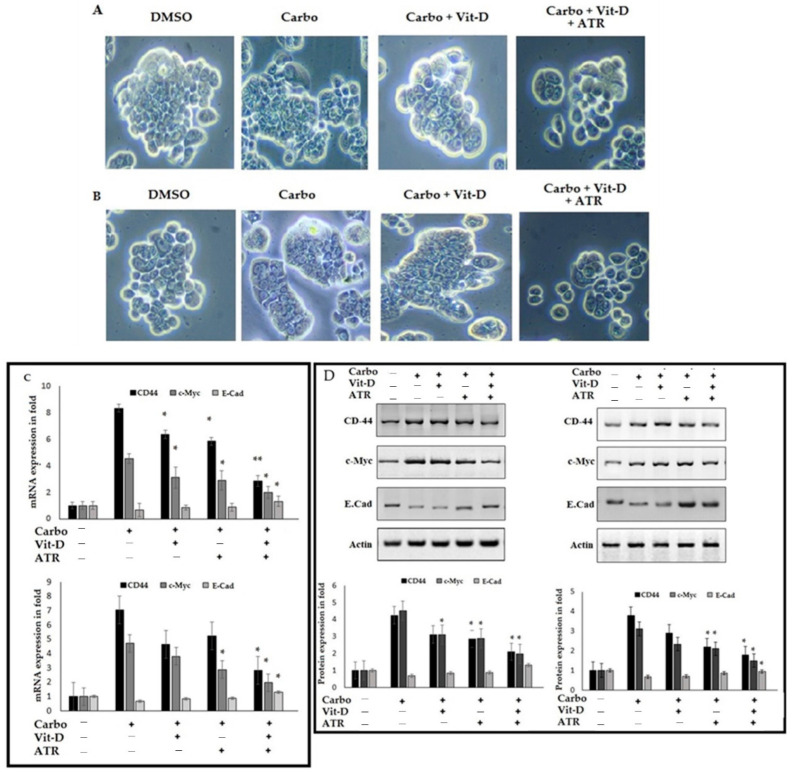
Effect of atropine and vit D on carboplatin-treated estrogen-sensitive and -insensitive breast cancer cells. (**A**,**B**) MDA-MB-231 and T47D cells were cultured in 12-well plate, induced by carboplatin (5 nM) and treated with or without vit-D-β (10 nm) supplemented with atropine (25 μmol/L) treatments. Control cells were supplemented with 1% DMSO. The stem cell colony morphology was recorded after 4 days of treatment. The red arrow mark indicates the sphere formation, loss, gain of its epithelial structure morphology in carboplatin alone, and atropine treatment (**A**,**B**) The colony formation can be compared with control cell. The inhibitory effect of atropine and vit D of E-cad, CD-44, and c-Myc gene (**C**) and protein expression (**D**) of carboplatin-treated cancerous cell lines. * *p* < 0.05 and ** *p* < 0.01 (one-way ANOVA).

**Table 1 ijms-23-09849-t001:** Interactions of atropine with E-cad and ZEB-2 residues of target protein.

Interaction with	Binding Energy	Ligand Efficiency	Intermole Energy	Ligand Atoms (Ring)	Docked Amino Acid Residue (Bond Length)
E-cad	−6.75	−0.32	−7.74	Conventional hydrogen bond: C12-H Pi-Alkyl hydrophobic bond: O	Chain A: GLN’197’OE1’ (2.01 Å) Chain A: LEU’196’ (4.53 Å)
ZEB-2	−4.78	−0.23	−5.85	Conventional hydrogen bond: C12-H Carbon hydrogen bond interaction C8 C12-O Alkyl hydrophobic bond: O O Pi-Alkyl hydrophobic bond: O O	Chain A: VAL’46’O’(1.75 Å) Chain A: PRO’10’O’ (3.09 Å) Chain A: LYS’49’CE’ (3.72 Å) Chain A: LEU’15’ (5.38 Å) Chain A: LEU’15’ (5.26 Å) Chain A: LEU’15’ (5.38 Å) Chain A: LYS’49’’CE’ (3.72 Å)

**Table 2 ijms-23-09849-t002:** List of used primers and their sequences.

Primer Name	Forward Primer	Reverse Primer	PCR Product Size
E-cad	GCCTCCTGAAAAGAGAGTGGAAG	TGGCAGTGTCTCTCCAAATCCG	189
CD-44	CCAGAAGGAACAGTGGTTTGGC	ACTGTCCTCTGGGCTTGGTGTT	212
ZEB2	AATGCACAGAGTGTGGCAAGGC	CTGCTGATGTGCGAACTGTAGG	231
c-MYC	CCTGGTGCTCCATGAGGAGAC	CAGACTCTGACCTTTTGCCAGG	168
SOX-2	GCTACAGCATGATGCAGGACCA	TCTGCGAGCTGGTCATGGAGTT	188
GAPDH	GTCTCCTCTGACTTCAACAGCG	ACCACCCTGTTGCTGTAGCCAA	195

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
