# Peer review of "Atropine Is a Suppressor of Epithelial–Mesenchymal Transition (EMT) That Reduces Stemness in Drug-Resistant Breast Cancer Cells"

_ijms, 2022, doi:10.3390/ijms23179849_

Round 1
Reviewer 1 Report
Dear Editor,
the manuscript entitled “Atropine is a Suppressor of Epithelial-Mesenchymal Transition (EMT), Reduces Stemness in Drug Resistant-Breast Cancer Cells” (Manuscript ID: ijms-1800374) describes the ability of atropine to reduce epithelial mesenchymal transition (EMT) and colony formation induced by TGF-B or carboplatin in both MDA-MB-231 and T47D cell lines, through the reduction of of CD-44 and C- Myc expression. Molecular modelling studies indicated that atropine is able to stably interact with E-Cadherin and Zeb2, two important signaling molecules involved in EMT regulation. Anti-invasive effects were also investigated.
The results of this study are quite interesting, but there are some issues that should be addressed before publication.
-The conclusion section should be improved, as it is too concise
-An extensive editing of English language and style are requested, since there are several sentences within the manuscript that should be reworded. Some examples are listed below:
· Pag. 7 lines 2-4: “These data indicate that atropine is a genotoxic drug reduces cell viability in breast cancer MDA‐MB‐231 and T47D cell lines”.
· Pag 11 last lines: “These data reflect a possible role for atropine when in EMT regulation used in combination with or alone.
· Pag.15, last lines: “Our results showed that the binding of atropine to E-cad and ZEB-2 and may help in reducing the breast cancer metastatic development and resistance to chemotherapy is two cell line from different origin the aggressive claudin-low mesenchymal-like cells and the T47D cells, is a luminal A epithelial like cell line.
· Pag.16, lines 4-8: “In this regard, M3-mAChRs were found to involve in tumor cell migration, perineural invasion, and EMT during cholangiocarcinoma [22]. However, in a recent study, an anticancer activity has been shown for the alkaloid atropine isolated from hyoscyamus albus plant among its methanolic on different cancerous cell lines [23].
· Pag.16, line 42: “Herein, atropine papers to induce a similar effect to what is induced by vit D”.
· Pag. 11, paragraph 3.5, lines 11-12: please remove “atropine papers to induce a similar expression level of E-cad to what is induced after vit D treatment.”
-Table 2 is not mentioned in the section of molecular modelling studies
-The authors should report the cited references in a uniform style. Moreover, reference 10 refers to 2 different citations (doi: 10.1007/s11684-018-0656-6 and 10.1016/j.semcancer.2012.04.001) and in reference 19 pages of the cited article are missing
-The quality of figure 1 should be improved
Other revisions are listed below:
· Authors’ affiliations are identified by a superscript number, unlike their names where letters are used.
· Please replace full stop with comma in reference ranges (see for example pag 2, [14.15] or [18.19], pag. 16 [2. 3].
· Pag. 3, paragraph 2.6, line 4: please remove the full stop after the bracket
· In paragraph 3.7, please check punctuation and font size
· Pag. 2, line 27: does 1d50 stand for ID50?
Author Response
We would like to express our gratitude to for time and constructive comments which have helped us to improve our contribution.
Below we proved detailed answers (marked in blue) to the referee’s comments.
The manuscript entitled “Atropine is a Suppressor of Epithelial-Mesenchymal Transition (EMT), Reduces Stemness in Drug Resistant-Breast Cancer Cells” (Manuscript ID: ijms-1800374) describes the ability of atropine to reduce epithelial mesenchymal transition (EMT) and colony formation induced by TGF-B or carboplatin in both MDA-MB-231 and T47D cell lines, through the reduction of of CD-44 and C- Myc expression. Molecular modelling studies indicated that atropine is able to stably interact with E-Cadherin and Zeb2, two important signaling molecules involved in EMT regulation. Anti-invasive effects were also investigated.
The results of this study are quite interesting, but there are some issues that should be addressed before publication.
-The conclusion section should be improved, as it is too concise
Response: The conclusion section has been improved and the conclusive statements were re-written.
-An extensive editing of English language and style are requested, since there are several sentences within the manuscript that should be reworded. Some examples are listed below:
- Pag. 7 lines 2-4: “These data indicate that atropine is a genotoxic drug reduces cell viability in breast cancer MDA‐MB‐231 and T47D cell lines”.
Response: This has been rephrased and corrected
- Pag 11 last lines: “These data reflect a possible role for atropine when in EMT regulation used in combination with or alone.
Response: This has been corrected
- Pag.15, last lines: “Our results showed that the binding of atropine to E-cad and ZEB-2 and may help in reducing the breast cancer metastatic development and resistance to chemotherapy is two cell line from different origin the aggressive claudin-low mesenchymal-like cells and the T47D cells, is a luminal A epithelial like cell line.
Response: Sentences have been shortened and corrected
- Pag.16, lines 4-8: “In this regard, M3-mAChRs were found to involve in tumor cell migration, perineural invasion, and EMT during cholangiocarcinoma [22]. However, in a recent study, an anticancer activity has been shown for the alkaloid atropine isolated from hyoscyamus albus plant among its methanolic on different cancerous cell lines [23].
Response: This has been corrected
Pag.16, line 42: “Herein, atropine papers to induce a similar effect to what is induced by vit D”.
Response: This has been corrected
Pag. 11, paragraph 3.5, lines 11-12: please remove “atropine papers to induce a similar expression level of E-cad to what is induced after vit D treatment.”
Response: This has been corrected
-Table 2 is not mentioned in the section of molecular modelling studies
Response: Table 2 has been now mentioned.
-The authors should report the cited references in a uniform style. Moreover, reference 10 refers to 2 different citations (doi: 10.1007/s11684-018-0656-6 and 10.1016/j.semcancer.2012.04.001) and in reference 19 pages of the cited article are missing
Response: Thank you, these references have been corrected and cited properly
-The quality of figure 1 should be improved
Response: Figure 1 has been provided in a better quality
Other revisions are listed below:
- Authors’ affiliations are identified by a superscript number, unlike their names where letters are used.
Response: That has been corrected
- Please replace full stop with comma in reference ranges (see for example pag 2, [14.15] or [18.19], pag. 16 [2. 3].
Response: That has been corrected
- Pag. 3, paragraph 2.6, line 4: please remove the full stop after the bracket
Response: That has been corrected
- In paragraph 3.7, please check punctuation and font size
Response: That has been corrected
- Pag. 2, line 27: does 1d50 stand for ID50?
Response: That has been corrected
Reviewer 2 Report
In this paper the authors aim to investigate the effect of atropine on the development of EMT in breast cancer. They find that in cells previously treated with TGFb, atropine inhibits EMT and reduces the expression of some effectors such as CD44 and Myc, as well as having a blocking effect on migration and invasiveness in the presence of EMT inducers such as carboplatin. The topic of study is interesting and relevant to the scientific community, as well as novel, however, I find serious flaws that incline me to reject the work. On the one hand, the cellular model of the study is not well explained in M&Ms. According to the figure captions and some parts of the text, it seems that the breast cancer cell lines MDA-MB-231 and T47D have been previously treated with TGFb, I assume to induce EMT, but it is not explained in M&Ms, which makes it difficult to understand the experimental design. In addition, there is contradictory information on the epithelial or mesenchymal phenotype of the two lines in the introduction which should be clarified. On the other hand, no comparison of the effect of atropine in the presence and absence of TGFb is made as to whether its effect on EMT is dependent on the pathways activated by TGFb or not. In addition, using as hypothesis that the fact that AchRs are therapeutic targets in breast cancer because they modulate EMT, but then studying the direct interaction of atropine to EMT factors makes little sense. Moreover, the quality of the figures is poor and makes them difficult to understand, parts of some figures are missing and titles on the axes of others are missing (check Figure 3 thoroughly), no exact IC50 values for atropine are given, etc. In conclusion, despite the interest that the subject of the study may have for the scientific community, the presentation and explanation of the results must be considerably improved.
Author Response
In this paper the authors aim to investigate the effect of atropine on the development of EMT in breast cancer. They find that in cells previously treated with TGFb, atropine inhibits EMT and reduces the expression of some effectors such as CD44 and Myc, as well as having a blocking effect on migration and invasiveness in the presence of EMT inducers such as carboplatin.
The topic of study is interesting and relevant to the scientific community, as well as novel, however, I find serious flaws that incline me to reject the work. On the one hand, the cellular model of the study is not well explained in M&Ms.
Response: Thank you very much for time and effort and comments, we have revised materials and methods to improve and explain the methodology
According to the figure captions and some parts of the text, it seems that the breast cancer cell lines MDA-MB-231 and T47D have been previously treated with TGFb, I assume to induce EMT, but it is not explained in M&Ms, which makes it difficult to understand the experimental design.
Response: This point has been clarified at the Materials and Methods section.
In addition, there is contradictory information on the epithelial or mesenchymal phenotype of the two lines in the introduction which should be clarified.
Response: We have revised and rephrased the sentences about the information about both of epithelial or mesenchymal phenotype of the two lines in the introduction to clarified
On the other hand, no comparison of the effect of atropine in the presence and absence of TGFb is made as to whether its effect on EMT is dependent on the pathways activated by TGFb or not.
Response: We already have done the comparison between the atropine treated TGF-B and the atropine treated DMSO cells, western blot data of the draft of the figure including that is now supplemented. Actually, since we could not see significant variations, we removed ATR/DMSO from the submitted figures to show simple and better quality graph have more space for the treated groups.
In addition, using as hypothesis that the fact that AchRs are therapeutic targets in breast cancer because they modulate EMT, but then studying the direct interaction of atropine to EMT factors makes little sense.
Response: For the hypothesis that atropine inhibits AchRs and therefore modulates EMT, we have presented what we have seen in that model using two different cell lines in which obvious dysregulation of EMT signaling molecules was found based on both western blot and PCR data. However, sill virtual interaction between atropine and E-cad and ZEB2 is supportive, we did not go deeper at that point, although we planned to do CoIP. Besides it is interesting to know that atropine indices a similar effect of vit D or even can be used in combination with vit D and other classical chemotherapeutic drugs to prevent breast tumor progression.
Moreover, the quality of the figures is poor and makes them difficult to understand, parts of some figures are missing and titles on the axes of others are missing (check Figure 3 thoroughly),
Response: We have updated figures and we do trust that the journal will adjust the missing parts in the next version.
no exact IC50 values for atropine are given, etc.
Response: We have revised that
In conclusion, despite the interest that the subject of the study may have for the scientific community, the presentation and explanation of the results must be considerably improved.
Response: Thank you for your comments, we have considered that all are supportive and we did the best to improve and explain the proposed point of the current research.
Round 2
Reviewer 2 Report
I found that the authors have made an extensive revision of manuscript and improved the the explanation of the methods and results, which was the weakest point of the article and which made it very difficult to evaluate in some sections. However, I still found minor mistakes in the wording and presentation of the figures that should be revised:
- the protocol followed to study the effect of vitamin D in EMT and migration/invasion is not explained in M&Ms.
- Figure 2: revise the spelling of the following sentence of the footnote "Normal breast cells, MDA‐MB‐231 and T47D treated with TGF-β (10 ng/ml) or a vehicle control (0.1% DMSO) cell viability was quantified after incubating cells with atropine for 48 hours for normal (A) cells and at time points 24, 48 and 72 hr for breast cancer cells (B)". Also, include description of part D of this figure.
-Figure 4B: specify what the numbers 3-50 refer to. I suppose it is atropine concentration. The same with parts D: specify what 10ng/ml and 10u is (TGF-b and atropine, I suppose).
- Figure 5: it is not stated which cell line the left and right results correspond to.
In Results, part 3.5, the description of whether cells are epithelial or mesenchymal has been again reversed: "As shown in Fig 5B, atropine diminishes the action of TGF-β inducing EMT in the epithelial like BC cells MDA-MB231 via down-regulating ZEB-2. In the mesenchymal like T47D cell line,...". Please, correct it.
-
Author Response
Comment: I found that the authors have made an extensive revision of manuscript and improved the the explanation of the methods and results, which was the weakest point of the article and which made it very difficult to evaluate in some sections. However, I still found minor mistakes in the wording and presentation of the figures that should be revised:
* Mistakes in wording have been corrected and the presentation of figures has been revised.
- the protocol followed to study the effect of vitamin D in EMT and migration/invasion is not explained in M&Ms.
- Answer: The protocol of the effect of vitamin D in EMT and migration/invasion has been now explained at M&Ms
- Figure 2: revise the spelling of the following sentence of the footnote "Normal breast cells, MDA‐MB‐231 and T47D treated with TGF-β (10 ng/ml) or a vehicle control (0.1% DMSO) cell viability was quantified after incubating cells with atropine for 48 hours for normal (A) cells and at time points 24, 48 and 72 hr for breast cancer cells (B)".
- Answer: That has been revised
- Also, include description of part D of this figure.
- Answer: Part D has been described.
-Figure 4B: specify what the numbers 3-50 refer to. I suppose it is atropine concentration. The same with parts D: specify what 10ng/ml and 10u is (TGF-b and atropine, I suppose).
* That has been clarified
- Figure 5: it is not stated which cell line the left and right results correspond to.
- Answer: That has been stated
In Results, part 3.5, the description of whether cells are epithelial or mesenchymal has been again reversed: "As shown in Fig 5B, atropine diminishes the action of TGF-β inducing EMT in the epithelial like BC cells MDA-MB231 via down-regulating ZEB-2. In the mesenchymal like T47D cell line,...". Please, correct it.
* Answer: That has been now corrected